# The Latest Advancements in Diagnostic Role of Endosonography of Pancreatic Lesions

**DOI:** 10.3390/jcm12144630

**Published:** 2023-07-12

**Authors:** Jagoda Oliwia Rogowska, Łukasz Durko, Ewa Malecka-Wojciesko

**Affiliations:** Department of Digestive Tract Diseases, Medical University of Lodz, 90-647 Lodz, Poland; lukasz.durko@umed.lodz.pl (Ł.D.); ewa.malecka-panas@umed.lodz.pl (E.M.-W.)

**Keywords:** endoscopic ultrasound, endoscopic ultrasound guided biopsy, pancreas, review

## Abstract

Endosonography, a minimally invasive imaging technique, has revolutionized the diagnosis and management of pancreatic diseases. This comprehensive review highlights the latest advancements in endosonography of the pancreas, focusing on key technological developments, procedural techniques, clinical applications and additional techniques, which include real-time elastography endoscopic ultrasound, contrast-enhanced-EUS, EUS-guided fine-needle aspiration or EUS-guided fine-needle biopsy. EUS is well established for T-staging and N-staging of pancreaticobiliary malignancies, for pancreatic cyst discovery, for identifying subepithelial lesions (SEL), for differentiation of benign pancreaticobiliary disorders or for acquisition of tissue by EUS-guided fine-needle aspiration or EUS-guided fine-needle biopsy. This review briefly describes principles and application of EUS and its related techniques.

## 1. Introduction

The origin of endoscopic ultrasound (EUS) dates to 1980, when images were obtained in dogs according to DiMagno et al. [1]. Since then, there has been significant progression in the diagnostic role of EUS, not only in its wide range of usage in pancreatic pathology but also in its safety [1]. Due to its innovative mechanism, which combines fibre-optic endoscopic and ultrasonic capabilities, EUS has become an enormously indispensable diagnostic tool for differentiation of subepithelial lesions [2]. What is more, a review of 66 studies showed that EUS was the most sensitive and specific investigation technique in identifying subepithelial lesions (SEL) < 2 cm compared to other imaging modalities such as computed tomography (CT) or magnetic resonance imaging (MRI), the sensitivities of EUS, CT and MRI were 93%, 53% and 67%, respectively [3,4].

Numerous studies have shown high sensitivity (92–100%), specificity (89–100%) and accuracy (86–99%) of EUS in the detection of pancreatic malignancies, which is higher than that of CT scan, particularly with small diameter lesions (Table 1) [5].

The increasing incidence of incidental pancreatic lesions has prompted a focus on their accurate diagnosis. These lesions, referred to as focal pancreatic lesions, can manifest as solid, cystic or mixed tumours. Solid lesions encompass a spectrum from benign (serous pancreatic cystadenoma, papillary cysts, lymphoepithelial cysts) to precancerous (intraductal papillary mucinous neoplasm (IPMN) with low-grade dysplasia, mucinous cystic neoplasm (MCN) with low-grade dysplasia, benign neuroendocrine tumours) and malignant (ductal adenocarcinoma, acinar cell carcinoma, IPMN with invasive carcinoma, cystadenocarcinoma, neuroendocrine tumours) [6,7]. Comparative diagnostic studies have evaluated various imaging techniques for characterizing focal pancreatic lesions, as presented in Table 2 of the multi-centre study by Best LM et al. [8].

Furthermore, there is an urgent need for advanced tools that not only aid in diagnosis but also facilitate tissue acquisition, forming the basis for therapeutic procedures [2]. EUS has introduced related techniques such as contrast-enhanced EUS, EUS elastography and EUS-guided fine-needle aspiration biopsy (EUS-FNA) for tissue sampling [9]. Contrast-enhanced EUS employs contrast agents to enhance visualization of blood flow within pancreatic lesions, assisting in the differentiation of malignant and benign lesions. EUS elastography provides information about tissue stiffness, aiding in the characterization of solid and cystic lesions [10]. EUS-FNA enables real-time ultrasound-guided sampling of tissue for histological and cytological analysis, enabling a definitive diagnosis [11].

Once a diagnosis is established, the treatment approach for focal pancreatic lesions varies depending on the nature of the lesion. Benign and precancerous lesions may be managed conservatively with regular monitoring, while malignant lesions often require intervention. Treatment options encompass surgical resection, endoscopic resection, ablation techniques and systemic therapies tailored to the specific diagnosis and disease stage.

To sum up, the diagnosis of focal pancreatic lesions is crucial, and various imaging techniques have been compared for their diagnostic capabilities. EUS-related techniques, such as contrast-enhanced EUS, EUS elastography and EUS-FNA, offer valuable tools for both diagnosis and tissue acquisition. The treatment strategy depends on the nature of the lesion, ranging from conservative management to invasive interventions, ensuring personalized care for patients with focal pancreatic lesions. 

**Table 1 jcm-12-04630-t001:** Studies on diagnostic performance of EUS versus CT for detection of pancreatic malignancy.

Study, Year	Cases	Sensitivity, EUS vs. CT (%)	Specificity, EUS vs. CT (%)
Due et al., 2017 [12]	68	98 vs. 73	NA
Kamata et al., 2014 [13]	35	100 vs. 56	100 vs. 100
Kitana et al., 2012 [14]	277	91 vs. 71	94 vs. 92

CT—computed tomography, EUS—endoscopic ultrasound, NA—not applicable.

**Table 2 jcm-12-04630-t002:** Different imaging techniques in characterizing pancreatic focal lesions according to Best et al. [8].

Diagnostic Technique	Cases	Sensitivity	Specificity	Post-Test Probability of Positive Test	Post-Test Probability of Negative Test
PET	99	92%	65%	86%	22%
EUS	133	95%	53%	82%	18%
EUS-FNA (cytology)	147	79%	100%	99%	32%
CT	123	98%	76%	90%	6%
MRI	29	80%	89%	94%	34%

PET—positron emission tomography, EUS—endoscopic ultrasound, FNA—fine-needle aspiration, CT—computed tomography, MRI—magnetic resonance imaging.

## 2. Principles of EUS-Related Techniques

### 2.1. Real-Time Elastography EUS (RTE-EUS)

Elastography is an imaging technique based on the evaluation of tissue stiffness, which leads to a better classification of lesions [15,16]. The principle of this method is explained by using the spring model. Under compression, hard springs are remotely deformed while soft springs compress significantly [16]. Malignant tumours are harder than benign ones [17]. There are two semi-quantitative elastography methods: SH (mean strain histograms) and strain ratio (SR) [18]. The mean strain histogram value corelates with the hardness of the lesion depicted by the colour on the scale from hardest (0) to softest (255). The system is set up to use a colour map (red-green-blue), where hard tissue areas appear as dark blue, medium hard tissue areas as cyan, intermediate tissue areas as green, medium soft tissue areas as yellow and soft tissue areas as red [15]. Elastography imaging of the normal pancreas is characterized by a homogenous green colour distribution (representing intermediate stiffness) [15]. Neuroendocrine tumours tend to be stiffer when compared to the pancreatic parenchyma, especially if they are malignant. When it comes to acute pancreatitis, the necrotic zones appear softer as compared to the stiffer surroundings [19]. In initial stages of chronic pancreatitis, a honeycomb pattern dominated by hard strands is reflected in elastography images [20].

Shear-wave elastography (SWE) is a newly introduced imaging technique that allows quantification of mechanical and elastic tissue properties. SWE uses an acoustic radiation force pulse sequence to generate shear waves, which propagate perpendicular to the ultrasound beam, causing transient displacements. What is more, shear waves propagate faster through stiffer contracted tissue [21]. SWE is able to assess the biomechanical properties of tissue; generally, malignant lesions are stiffer than the healthy parenchyma. Principal applications are determination of fibrosis and autoimmune pancreatic diseases, characterization of pancreatic lesions, guiding biopsy in the stiff part of a focal area or characterization of pancreatic gland stiffness in suspected chronic pancreatitis [22].

Giovannini et al., in a multi-centre study including 121 patients, demonstrated that the sensitivity and specificity of EUS elastography for malignancy in pancreas were 92.3% and 80%, respectively.

### 2.2. Contrast-Enhanced-EUS

CE-EUS is a remotely new established diagnostic examination that contains both high-resolution ultrasound and the administration of ultrasound contrast agents [23]. The technique was invented by using two different methods: contrast-enhanced endoscopic Doppler ultrasound with a high-mechanical index (CEHMI-EUS) (this one does not require special software) or the second one, which runs on the specific mode, contrast-enhanced low-MI EUS (CELMI-EUS) [23]. The introduction of contrast enhancers could provide additional information about the vascularization of the organ, which resulted in increased value of the method, especially for diagnosing necrotic pancreatic areas [23]. There are three contrast agents that are currently available: sulphur hexafluoride (SF6) gas with a lipid stabilizer shell, octafluoropropane (C3F8) with a lipid stabilizer shell or perfluorobutane (C4F10) with a lipid stabilizer shell; the last one is not available in Europe, apart from Norway and Denmark [24,25]. When the agents are administered through a peripheral vein, the microbubbles in the contrast agent receive transmitted US waves and are disrupted or stimulated to resonate, thereby producing the signal detected in the US image, which has low interferences [4]. The main elements and advantages of CE-EUS include real-time imaging of microvascularity and microperfusion, real-time intervention guidance, on-site performance ability and impressively good detail resolution [23].

Clinical applications include differential diagnosis of focal pancreatic masses and evaluation of acute and chronic pancreatitis, particularly complications associated with pancreatitis, assessment of cystic lesions, characterization of intraductal biliary/pancreatic structures gallbladder lesions, SEL, lymph node assessment and others [26,27]. Kamata et al. [13] reported that CE-EUS identified mural nodules more accurately than conventional EUS, providing sensitivity and specificity values of 97% and 75% for CE-EUS and 97% and 40% for conventional EUS. This differentiation between mural nodules and mucous clots is crucial to distinguish MCNs from IPMN [4].

CE-EUS is believed to be beneficial in differentiating pancreatic adenocarcinoma and neuroendocrine tumours [22]. According to Ishikawa T et al. [28] CE-EUS has been reported to have a high sensitivity in identifying PNETs compared to CT with values of 95% and 81%, respectively. What is more, CE-EUS detects a heterogeneous tumour texture, which is a significant sign of malignancy [29]. Due to the study conducted by Leem G et al. [30], CEH-EUS of the pancreatic solid masses showed higher sensitivity and specificity in differentiating pancreatic adenocarcinoma and neuroendocrine tumours (82.0% and 87.9% for pancreatic adenocarcinoma and 81.1% and 90.9% for neuroendocrine tumours, respectively) [13].

Less than 5% of pancreatic masses represent metastases and their differentiation from primary tumours using conventional EUS is difficult [31]. CH-EUS, due to its ability to provide information about the vascularization of the organ, can detect pancreatic metastatic lesions, which according to Teodorescu C et al. [31] are mostly hypervascular. These metastases can have a hyperenhanced aspect (renal cell carcinoma or melanoma) or a hypovascular aspect (colon cancer, breast carcinoma).

What is more CE-EUS has been reported to have better accuracy than contrast-enhanced multidetector CT (MDCT) for early diagnosis of small pancreatic cancer. In the Japanese study from 2020, the sensitivity of CE-EUS and MDCT was 91.2% and 70.6%, respectively [27,32]. CE-EUS was also significantly more accurate than the standard EUS in diagnosing malignant cysts with accuracy (84% vs. 64%).

### 2.3. EUS-Guided Fine-Needle Aspiration (EUS-FNA)

This technique has been used as a gastroenterological standard for sampling pancreatic solid masses, SEL and lymph nodes since 1992 [33,34]. When the aspirate is sufficient for cytology, it’s accuracy ranges from 77% to 95% for pancreatic masses [35,36]. In general, 19 G–25 G calibre needles are inserted under EUS guidance for the pathological diagnosis of pancreatic cancer and lymph nodes [4]. EUS-FNA is often performed in the evaluation of pancreatic cystic lesions (PCL) for a better preoperative characterization [37]. One of the most crucial limitations of EUS-FNA is the fact that it does not provide core tissue specimens with preserved architecture, therefore immunohistochemical staining and histologic diagnosis cannot be assessed [38]. Rapid on-site evaluation (ROSE) refers to the immediate cytologic assessment after FNA by a cytopathologist, and is useful for increasing the accuracy and sample acquisition and reducing the number of needle passes in EUS-FNA [39]. Observational studies demonstrated that ROSE improved the diagnostic accuracy and tissue adequacy of EUS-FNA, particularly in solid pancreatic lesions. However, four meta-analyses suggested a modest improvement in sensitivity with ROSE, but the difference was not statistically significant, while other meta-analyses did not support the advantages of ROSE in terms of specimen adequacy and diagnostic yield [40]. Therefore, The European Society of Gastrointestinal Endoscopy panel recommends EUS-FNA with or without ROSE equally, given the conflicting evidence [41].

### 2.4. EUS-Guided Fine-Needle Biopsy (EUS-FNB)

In order to overcome the limitations of EUS-FNA, in the early 2000s, EUS-FNB was introduced to obtain a tissue specimen and a molecular analysis [33]. The pooled data showed EUS-guided pancreas biopsy could be a safe approach for the diagnosis of pancreatic tumours [4]. EUS-guided fine-needle biopsy uses a Franseen needle to sample considerable material with a small number of punctures [34].

Table 3 presents superiority of EUS-FNB over EUS-FNA in the diagnosis of pancreatic cancer, however, they are equivalent when it comes to detecting SEL [34].

Although EUS-guided tissue acquisition is a standard modality for establishing a conclusive diagnosis and individualized therapeutic plan for pancreatic solid tumours, the diagnostic performance has been reported to have a wide range according to the needle type [44,45].

Based on the systematic review and network meta-analysis [46] conducted on the comparative diagnostic performance of end-cutting fine-needle biopsy (FNB) needles for tissue sampling of pancreatic masses, several key findings emerged. Franseen needles and Fork-tip needles exhibited superior diagnostic accuracy and sample adequacy compared to reverse-bevel needles and FNA needles. Among the different needle sizes, 25-gauge Franseen and Fork-tip needles did not show superiority over 22-gauge reverse-bevel needles. Importantly, when rapid onsite cytologic evaluation was available, none of the tested FNB needles demonstrated significant superiority over other FNB devices or FNA needles. Overall, Franseen and Fork-tip needles, particularly in 22-gauge size, showed the highest performance for tissue sampling of pancreatic masses. However, it is essential to note that the confidence in these estimates was low, underscoring the need for further research and validation in this field [47].

### 2.5. EUS-Guided Rendezvous Technique (EUS-RV)

The endoscopic ultrasound-guided rendezvous technique (EUS-RV) is a promising procedure used in gastrointestinal endoscopy when conventional methods like endoscopic retrograde cholangiopancreatography (ERCP) are not successful. EUS-RV combines EUS imaging with therapeutic intervention to achieve access to the biliary system [48].

By creating a connection between the biliary and gastrointestinal tracts, EUS-RV enables successful biliary access for diagnostic and therapeutic purposes. Using a specialized linear array echoendoscope, this technique provides high-resolution ultrasound imaging and targeted interventions. EUS-RV overcomes anatomical challenges and offers a less invasive option for patients who may not be suitable candidates for ERCP [49].

The procedure involves inserting the echoendoscope, visualizing the biliary tree and puncturing the gastrointestinal wall under ultrasound guidance. A guidewire is then placed into the bile duct, creating a pathway for subsequent interventions. EUS-RV has shown success in cases of difficult biliary cannulation, altered anatomy and previous surgeries [49].

EUS-RV is a valuable technique used in the management of pancreatic ascites resulting from pancreatic duct (PD) leaks. While PD disruption and resultant ascites are more commonly associated with chronic pancreatitis, it is rare in cases of acute necrotizing pancreatitis. Medical therapy, surgical management or endotherapy are the available options for managing pancreatic ascites [50,51].

Factors that contribute to a successful EUS-RV procedure have been identified, with a dilated PD being essential for optimal outcomes. However, the literature lacks reports on EUS-guided rendezvous in a nondilated PD. The procedure offers a potential solution for cases where ERCP fails to achieve selective cannulation, allowing for successful access to the PD and subsequent endotherapy [50,51].

EUS-RV has shown promise as an effective technique for managing pancreatic ascites associated with PD leaks. By utilizing the capabilities of EUS, this procedure provides a minimally invasive approach to accessing the PD and facilitating appropriate intervention. Further research and clinical experience are necessary to refine the technique and identify the optimal patient selection criteria for successful EUS-RV in both dilated and nondilated PDs [50].

## 3. EUS in Pancreatic Pathologies

### 3.1. Pancreatic Cancer

Pancreatic cancer is currently the seventh leading cause of cancer death worldwide [52]. The five-year survival rate is exceptionally low—less than 10% [53]. Unfortunately, due to the huge progress in surgery, most of the cases are diagnosed at an advanced stage, so as a result only few patients can be qualified for surgery. What is more, this kind of treatment is still associated with high post-operative morbidity [52]. The commonly used term “pancreatic cancer” usually refers to ductal adenocarcinoma (PDAC), which represents 85% of all pancreatic tumours. Despite the ongoing developments, surgery is still associated with high post-operative morbidity [54].

EUS is considered to be the most sensitive technique to detect early neoplasia in the pancreas, which is presented as a hypoechoic mass with irregular borders. Typically, the dilatation of the proximal PD occurs. Unfortunately, when it comes to evaluation of distant metastasis, CT is superior to EUS [54]. The most widely used technique for the initial evaluation is the CT scan, with a sensitivity between 76% and 92%. Nonetheless, the sensitivity of EUS in detecting pancreatic lesions is around 98%, therefore it is the most sensitive technique for the detection of small pancreatic tumours [55]. Maguchi et al. [56] compared different imaging techniques for pancreatic cancer with a diameter < 2 cm and found that transabdominal ultrasound, CT and EUS had a sensitivity of 52.4%, 42.8% and 95.2%, respectively. What is more, EUS is superior to conventional imaging techniques due to such advantages as the lack of dosing ionizing radiation to the patient and the absence of contraindications such as metal implants or claustrophobia [57].

Since the EUS was invented, it has been also used to visualize a pancreas mass directly, secure a definitive cytologic or histologic diagnosis, define the degree of tumour-vascular involvement and more [58].

Conventional EUS functions may be enhanced by the newer related technique—EUS elastography. For instance, to obtain a histologic diagnosis and to provide material for molecular testing, EUS elastography can be merged with EUS-FNB in order to guide the biopsy. What is more, one of the recent prospective single-centre studies showed that EUS-FNA had a sensitivity, specificity and accuracy of 77.8%, 100% and 84% for pancreatic cancer diagnosis, respectively [59]. However, there is a possibility to achieve a higher diagnostic rate by combining real-time tissue elastography (RTE) with EUS-FNA, which was reported to have diagnostic accuracy, sensitivity and specificity of 94.4%, 93.4% and 100%, respectively [60].

According to recent clinical research, EUS-elastography, based on mean strain histogram and mass elasticity, is able to distinguish benign from malignant pancreatic tumours with a high sensitivity (Table 4) [15,61]. During EUS elastography, one trapezoidal region of interest (ROI) containing at least 50% of the lesion is manually selected. To calculate SH, a smaller round ROI is selected at the level of the focal lesion without the need to include a reference area [18]. The mean SH value represents the overall hardness of a lesion, with lower values (<80) being predictive of malignancy and higher values (>80) predictive of benign lesions. Combined CE-EUS (where the lesion is hypovascular) and SH with a cut-off value of 80 have shown to be the most specific and sensitive diagnostic method (98.6% and 81.4%, respectively) for detecting pancreatic carcinoma, according to Costache MI et al. [18] the average sensitivity of mean SH values ranges from 85% to 96%, while specificity ranges from 64% to 76% in detecting pancreatic tumours [18].

Figure 1 presents pancreatic adenocarcinoma adjacent tdetected using EUS-elastography.

The results of a recent meta-analysis showed pooled estimates of sensitivity and specificity of CEH EUS for pancreatic cancer diagnosis at 93% and 80%, respectively [57].

Moreover, to discriminate tumour lesions from inflammatory pancreatic masses, contrast-enhanced EUS may be used. When it generates an acoustic signal, as mentioned above, it helps in the assessment of vascularity of pancreatic masses in addition to providing information about the echogenicity of the lesions [5]. Iso-enhancement or hypo-enhancement, arterial irregularity and absent venous vasculature within a mass are typical for pancreatic PDAC, whereas hyper-enhanced lesions with the preserved architecture point to chronic pancreatitis [64,65,66]. CE-EUS can differentiate pancreatitis from pancreatic cancer with sensitivity, specificity, positive predictive value and negative predictive values of 91%, 93%, 100% and 88%, respectively [5].

The differences in diagnostic abilities of RTE-EUS and CE-EUS are shown in Table 5.

### 3.2. Pancreatic Neuroendocrine Tumours

Pancreatic neuroendocrine tumours (PNETS) are 7–10% of all pancreatic solid lesions. A majority of them (50–60%) are not secreting NETS [14]. Neuroendocrine tumours are malignant lesions that arise from neuroendocrine cells. They mostly occur in the gastrointestinal tract (48%), lung (25%) and pancreas (9%) [67]. Among the secreting endocrine tumours affecting the pancreas, insulinomas and gastrinomas tend to be the most common [68].

The use of EUS in the diagnosis and localization of PNETs has become increasingly a routine procedure [69]. With EUS, PNETs can be found at a low grade, what translates to a prompt surgery and a higher survival rate. According to the study conducted by Fujimori N et al. [70], EUS showed significantly higher sensitivity (96.7%) for identifying PNETs than CT (85.2%), MRI (70.2%) and ultrasonography (75.5%). What is more, the sensitivity of EUS-FNA for the diagnosis of PNET was 89.2%. The smaller size of the tumour was (<2 cm) the higher the concordance between EUS-FNA and surgical specimens, which is 87.5% [70]. EUS findings can differentiate between G1 and G2/G3 PNETs, with G2/G3 tumours more likely to be larger in size (>20 mm), heterogeneous and associated with main pancreatic duct (MPD) obstruction. Large tumour diameter and MPD obstruction are significantly associated with G2/G3 tumours, indicating a more advanced grade. EUS and EUS-FNA are considered highly sensitive and accurate diagnostic methods for PNETs. Characteristic EUS findings, such as large tumour size and MPD obstruction, can help in the grading of PNETs, particularly identifying G2/G3 tumours. To conclude, EUS and EUS-FNA are valuable tools for the diagnosis and grading of PNETs, providing important clinical information for treatment planning and management decisions [70].

Hypervascularization is another feature typical for PNETs, which used in imaging studies. According to the recent research by Battistella A et al. [71], hypervascularization is a common characteristic of PNETs and aids in their identification during imaging studies. However, the density of microvessels within PNETs can vary depending on their biological behaviour, and lower microvessel density is associated with more aggressive disease. The study found that a low microvessel density is indicative of aggressive behaviour in patients with nonfunctioning PNETs. Additionally, contrast-enhanced CT and contrast-enhanced EUS were identified as reliable and readily accessible methods for preoperatively assessing microvessel density in these tumours [71].

According to Deguelte S et al. [72], EUS is the most sensitive examination for PNET diagnosis with a detection rate of 86%. It can detect PNETs smaller than 2 cm with a great specificity. Therefore, EUS became part of the surveillance protocol for patients with hereditary syndromes (such as MEN-1 syndrome—multiple endocrine neoplasia-1) [72]. What is substantial to mention is the fact that the main risk factor for metastases in MEN-1 is the pancreatic tumour size [73]. According to the study conducted by the Endocrine tumour group, EUS detected nearly 85% of PNETs larger than 1 cm, whereas MRI visualized only 67% of them [73]. In addition, in pancreatic solid tumours, EUS can be combined with trans-gastric or trans-duodenal FNB [72]. Due to their rich vascularization, PNETs typically enhance with contrast for all modalities of imaging with early arterial enhancement like CE-EUS [72].

EUS should play a part in preoperative assessment, especially when there is an indication to perform a pancreatic parenchyma-sparing surgery, because it can define the anatomic relationship of the PNET to the PD and vascular structures [5]. Assessment of the distance between the PNET and the MPD is important before considering enucleation (where a distance of 2–3 mm between the lesion and the duct is usually recommended to limit the risk of ductal deformation and postoperative pancreatic fistula) [72].

Figure 2 presents images obtained from EUS-elastography detecting neuroendocrine tumour with high hardness strain ratio; while Figure 3 NET of the pancreas, enhancing after contrast administration was detected.

### 3.3. Pancreatic Cysts

Nowadays, PCL has become commonly recognized with an increasing frequency, so that the detection rate is rising with the advances in imaging technology, and there is an increased incidence of detection of unsuspected small PCLs [37,74]. Cystic lesions of the pancreas are classified into simple retention cysts, pseudocysts and cystic tumours [75]. The most common pancreatic cystic tumours include the IPMN, MCN and serous cystic adenoma. Numerous international guidelines recommend the qualification for surgical treatment of patients with a pancreatic cyst with a diameter of more than 30 mm, with the presence of adjacent tissue masses and concomitant dilatation of the Wirsung duct. The presence of such changes is associated with an increased risk of malignant transformation [76].

#### 3.3.1. Intraductal Papillary Mucinous Neoplasm 

The European Study Group on Cystic Tumours of the Pancreas recommends EUS and MRI as a method to diagnose the type of PCL [66]. Additionally, CH-EUS might detect hyperenhancement of a mural nodule, solid mass or septations, which point to malignancy. EUS-fine-needle aspiration (FNA) helps to determine the type of PCL [66]. The aspirated cystic fluid can be assessed for carcinoembryonic antigen (CEA), amylase levels and cytology. These parameters have the highest accuracy for differential diagnosis of mucinous from non-mucinous PCNs [77].

Mutations in the GNAS gene play a significant role in the diagnosis of IPMN of the pancreas. The study conducted by Kadayifci A et al. [78] demonstrates that GNAS testing, in combination with KRAS and CEA testing, enhances the accuracy of diagnosing IPMN. The presence of a GNAS mutation is highly specific to IPMN, with 47.2% of IPMN patients showing a positive GNAS result. When GNAS testing is added to CEA and KRAS testing, the overall diagnostic accuracy significantly increases to 86.2%. However, while the addition of GNAS to CEA improves accuracy, it does not surpass the diagnostic superiority of KRAS testing alone. In conclusion, the GNAS mutation serves as a valuable molecular marker in distinguishing IPMN and its inclusion in testing panels enhances the accuracy of IPMN diagnosis [78].

Numerous guidelines have been published to provide clear diagnostic and therapeutic recommendations for IPMNs. One of them is the Fukuoka Guidelines, which have been the current diagnostic and treatment standard for these tumours since 2017 [79]. This classification distinguishes three different types of IPMNs:Fukuoka-positive IPMNs—that have high-risk stigmata for malignancy (localized in pancreatic head leading to obstructive icterus, with mural nodules 5 mm in size and with dilation of the MPD to 10 mm).IPMNs with Fukuoka “worrisome features” (clinical signs of pancreatitis, dilation of the MPD to 5–9 mm, increased serum CA 19-9 values, clinical signs of pancreatitis).Fukuoka-negative IPMNs—without high-risk stigmata and without the “worrisome features” described above.

The treatment recommendations were also included. In resectable tumours, Fukuoka-positive IPMNs should be treated surgically. “Worrisome features”, EUS signs of mural nodules 5 mm, evidence of ductal changes or cytology suspicious for malignancy or even malignancy should also be indication for surgery. If neither of these is true, CT/MRI or EUS studies should be performed at intervals depending on the size of the IPMN [79].

The American Gastroenterological Association (AGA) presented the official recommendations on the management of pancreatic cysts in 2015 [80]. The most crucial recommendation is that patients with pancreatic cysts measuring less than 3 cm, without a solid component or a dilated PD, should undergo MRI surveillance after one year and then every two years for a total of five years, as long as there are no changes in size or characteristics.

For pancreatic cysts with at least two high-risk features, such as a size of 3 cm or larger, a dilated MPD or the presence of an associated solid component, the recommendation is to perform an examination using EUS-FNA.

Patients who receive non-concerning results from EUS-FNA should continue with MRI surveillance after one year and then undergo subsequent MRIs every two years to ensure that there are no changes in the risk of malignancy.

In cases where patients have both a solid component and a dilated PD, along with concerning features on EUS and FNA, surgery is recommended to reduce the risk of mortality from carcinoma [74].

These guidelines provided by the AGA offer valuable guidance on the surveillance and management of pancreatic cysts, taking into account specific criteria and risk factors. It is important for clinicians to follow these recommendations in order to make well-informed decisions regarding the appropriate management strategy for patients with pancreatic cysts.

#### 3.3.2. EUS-FNA

The gold standard in differentiating PCLs is EUS-FNA, which enables the use of aspirated samples for cytopathology examination and biochemical analyses, which provide an opportunity to further enhance diagnosis and medical decision making [81]. However, some sonographic findings of PCLs are indicative of malignancy, including a thick wall, septations and the presence of mural nodules, unfortunately, sonographic appearance or cytopathological examination has still a low predictive value for its diagnosis [82]. Nevertheless, the meta-analysis study conducted by Wang QX et al. [83] found that the pooled sensitivity and specificity for malignant cytology were 51% and 94%, respectively.

The aim of the biopsy is to distinguish premalignant lesions from malignant and it is usually performed for better preoperative characterization of the lesion [37].

To sum up, EUS-FNA is a useful tool for the differential diagnosis of benign (mucinous) and probably malignant cysts (non-mutinous), which is clearly presented in Table 6.

One of the main conclusions of the study conducted by Rogart et al. [87] was the fact that cyst wall puncture and aspiration during routine EUS-FNA may be a safe and easily applied. In the study, among patients with CEA < 192 ng/mL, 31% showed positive cytology for mucinous epithelium when CWP was employed. Additionally, when CEA analysis was not feasible due to insufficient fluid, CWP identified positive cytology for mucinous epithelium in 47% of the cysts. This cumulative approach using CWP resulted in an additional diagnostic yield of 37% for mucinous cysts. These findings demonstrate that incorporating CWP into the diagnostic process enhances the detection and characterization of mucinous cysts. Moreover, EUS-FNA enables clinicians to perform molecular analysis of cyst fluid, like KRAS mutation analysis. The latter increased the diagnostic accuracy of IPMNs to 81% [87,88].

However, EUS-FNA plays a significant role in the assessment of pancreatic cyst histotype, it also carries a notable risk of adverse events (AEs). To better understand the predictors for TTNB-related AEs and develop a prognostic model, a multi-centre retrospective analysis was conducted on 506 patients with PCLs who underwent TTNB. The study found that age, the number of TTNB passes, complete aspiration of the cyst and a diagnosis of IPMN were independent predictors of AEs. These findings were validated through logistic regression and random forest analyses. A hierarchical risk classification system was generated, identifying highrisk (IPMN with multiple microforcep passes), low-risk (patients < 64 years with non-IPMN diagnosis, ≤2 microforcep passes and complete cyst aspiration) and middle-risk groups. The study concludes that TTNB should be used selectively in patients with IPMN, and the developed model can assist in optimizing the benefit–risk balance of TTNB by aiding in patient selection [89].

The new disposable Moray micro forceps biopsy (MFB) device allows tissue sampling from the pancreatic cyst wall/septum and aims to improve diagnosis [90]. Recent meta-analyses have demonstrated that this instrument can significantly enhance the diagnostic accuracy of tissue sampling in patients with PCLs. Due to its effectiveness, MFB has been suggested as a valuable tool for diagnosing, characterizing and stratifying PCL. This disposable micro forceps can be inserted through a 19-gauge needle to obtain tissue samples from the cyst wall and/or septations. It enables histological examination of the architectural features and subepithelial stroma [91].

In the current study conducted by Zhang ML et al. [90], the diagnostic performance of PCF analysis and MFB was found to be similar in terms of diagnostic yield, mucinous cyst diagnosis and detection of high-risk cysts, with both methods achieving a diagnostic yield of over 70%. However, MFB outperformed PCF analysis in diagnosing specific types of cysts. Notably, MFB allowed for the diagnosis of 2.7 times more specific cysts compared to PCF analysis, across all cysts and specifically among those measuring less than 3 cm in size. In summary, the MFB allow for pancreatic cystic sampling with a higher level of precision.

EUS-FNB has revolutionized sampling techniques for pancreatic and nonpancreatic lesions, enabling histological evaluation and immunohistochemical staining. Two popular techniques, slow-pull and wet-suction, have been compared to the standard suction method. Slow-pull involves gradual withdrawal of the stylet to create negative pressure, while wet-suction flushes the needle with saline and applies suction using a pre-vacuum syringe. Previous studies have focused on comparing the standard suction and slow-pull techniques, showing similar adequacy and accuracy, with lower blood contamination in slow-pull. However, the wet-suction technique, introduced more recently, has demonstrated promising results in terms of specimen cellularity, adequacy and accuracy for both pancreatic and nonpancreatic lesions [92].

The findings from the study conducted by Crinò SF et al. [92] indicate that wet-suction showed a higher rate of tissue core acquisition, particularly in nonpancreatic lesions, suggesting that suction during the biopsy procedure may improve tissue quantity. Although the slow-pull technique did not significantly differ in tissue core acquisition, it demonstrated lower blood contamination, which may impact histological evaluation. Other strategies, such as the use of macroscopic on-site evaluation (MOSE), were not evaluated in this study. The choice between wet-suction and slow-pull should consider lesion characteristics, pathologist preference and operator experience. Further research and evidence are needed to comprehensively assess and compare the efficacy, safety and diagnostic yield of different sampling strategies.

The study conducted by Mangiavillano B et al. [93] compared the use of MOSE during EUS-guided fine-needle biopsy (EUS-FNB) with conventional EUS-FNB with three needle passes for pancreatic masses. The results showed that MOSE was noninferior to conventional EUS-FNB in terms of diagnostic accuracy, sample adequacy and safety. MOSE reliably assessed sample adequacy and reduced the number of needle passes required for diagnosis using a 22G Franseen needle. Incorporating MOSE into tissue sampling strategies for pancreatic masses can potentially optimize the procedure by improving efficiency without compromising diagnostic accuracy. Further research is needed to validate the role and benefits of MOSE in pancreatic tissue sampling.

#### 3.3.3. Carcinoembryonic Antigen 

CEA is currently considered the most accurate marker for differentiating mucinous, from non-mucinous cysts. Initially, the accuracy of cystic fluid CEA has been superior to EUS, cytology or other tumour markers. The optimal cut-off for differentiating mucinous from non-mucinous cysts was identified to be 192 ng/mL, which was associated with 75% sensitivity, and 84% specificity. Nevertheless, the recent meta-analysis of 18 studies with 1438 patients proved that CEA has 63% sensitivity and 88% specificity for identifying mucinous cysts. Another issue with this marker involves obtaining sufficient cyst fluid to assess CEA levels, which is often not possible, particularly in exceedingly small cysts [94].

#### 3.3.4. Amylase

Its cyst fluid level can be useful in excluding a pseudocyst from other types of pancreatic cysts. A large meta-analysis [95] found that a level of < 250 IU/L had a remarkably high specificity of 98% for excluding a pseudocyst. According to Thornton GD et al. [96] cyst fluid amylase level is similar in IPMNs and MCNs, therefore, its elevated level cannot be used to differentiate these two types of cysts.

#### 3.3.5. Cytology

Cyst fluid for cytology usually has a low diagnostic yield and less than 50% sensitivity for mucosal lesions, but is useful if positive for a specific diagnosis. Similarly, cytology is highly specific for malignancy, with at best a 60% sensitivity for malignancy [97].

#### 3.3.6. Glucose

Cyst fluid glucose levels have shown promise as a valuable diagnostic marker for mucinous cysts and may be more accurate than CEA for mucinous cysts. Cyst fluid glucose levels tend to be significantly lower in mucinous cysts compared to non-mucinous cysts. A cut-off value of <50 mg/dL has been proposed to optimize diagnostic accuracy. According to a recent meta-analysis, glucose has a sensitivity of 91% and a specificity of 75%, while CEA has a sensitivity of 67% and a specificity of 80% [98].

#### 3.3.7. CH-EUS

The quantum leap in diagnosing PCLs’ malignancy is said to be CH-EUS. It has a better ability to detect mural nodules, which can be a sign of a malignant cyst. Its improved ability can be assigned to the injected second-generation ultrasound contrast agents, which can detect microcirculation with better resolution and fewer artefacts than Doppler EUS images [6]. According to a prospective Zhong L et al. [99] study on CE-EUS for differential diagnosis of PCL, CE-EUS demonstrated greater accuracy in identifying PCNs than did CT, MRI or EUS-FB (fundamental B-mode)—CE-EUS vs. CT: 92.3% vs. 76.9%; CE-EUS vs. MRI: 93.0% vs. 78.9%; CE-EUS vs. FB-EUS: 92.7% vs. 84.2%. In the study conducted by Ohno E et al. [100], MPD involvement was diagnosed using CH-EUS in 90 patients with a sensitivity, specificity and accuracy of 83.5%, 87.0% and 84.9%, respectively. These results favoured it enough to be recommended by The European Study Group on Cystic Tumours of the Pancreas for being considered for further evaluation of mural nodules and assessing vascularity within the cyst and septations [66].

#### 3.3.8. EUS-nCLE

Confocal laser endomicroscopy (CLE) is a newly developed endoscopic technique that enables both the endoscopist and the pathologist, real-time imaging of tissue and vascular microstructures [101,102]. In this examination, a 19 G EUS needle is used, in which the stylet is replaced by the confocal mini-probe [102]. According to research conducted by Napoleon B et al. [101], there are three ample pieces of clinical evidence for an added benefit of the application of nCLE to EUS-FNA in the management of PCLs:EUS-nCLE provides better differentiation of mucinous and non-mucinous PCLs compared to the current standard of care.EUS-nCLE can improve the accuracy of diagnosis of PCLs, therefore reducing the rate of unnecessary follow-up investigations or inappropriate resections.The interobserver agreement for EUS-nCLE to differentiate mucinous from non-mucinous PCLs is high.

According to Giovannini M [102], the presence of epithelial villous structures based on nCLE was associated with pancreatic cystic neoplasm (IPMN) and provided a sensitivity of 59%, specificity of 100%, positive predictive value of 100%, and negative predictive value of 50%. Although, these data suggested that nCLE has a high specificity in the detection of IPMN, it may be limited by a low sensitivity.

### 3.4. Autoimmune Pancreatitis

Autoimmune pancreatitis (AIP) is an inflammatory process of the pancreas with a presumed autoimmune ethology, which is regarded as a separate type of chronic pancreatitis [103]. Two distinct types of AIP have been identified: AIP type 1 (AIP-1), considered the pancreatic manifestation of an IgG, related multiorgan disease, and AIP type 2 and is characterized by lymphoplasmacytic sclerosing pancreatitis (LPSP), pancreatic swelling, PD narrowing, obliterative phlebitis and IgG4-positive plasma cell infiltration [104,105,106]. However, AIP-2 is considered as a pancreatic-specific disease unrelated to IgG and is characterized by idiopathic duct-centric chronic pancreatitis, which is histopathologically represented by granulocytic epithelial lesions [104,106]. Clinical presentation of AIP, such as obstructive jaundice, abdominal pain and weight loss, mimics misleadingly pancreatic cancer (PC). What is more, AIP can also cause peripancreatic lymphadenopathy and vascular invasion, which makes differentiating AIP from PC challenging [103].

Diagnostic criteria are based on imaging findings of the pancreatic parenchyma, serological findings and response to steroid therapy [75]. Although diagnostic criteria are very similar, however, the method for analysing each finding varies depending on the country. For instance, in Japan endoscopic retrograde pancreatography (ERCP) is performed, while, in contrast, in the United States pancreatic core biopsy is routine for diagnosing AIP [75]. However, there is a common consensus that histology is a key criterion for the diagnosis of AIP. According to Matsubayashi H et al. [75], IgG4 (≥135 mg/dL) is the most specific serum marker for type 1 AIP with 86% sensitivity and 96% specificity to AIP against PC. Nevertheless, IgG4 is not actually specific for AIP.

According to Ishikawa T et al. [107], EUS can reveal pancreatic parenchymal and ductal features in much more detail than any other existing imaging modality. However, differentiating AIP and PC based on hypoechoic masses using conventional EUS is difficult, there may be some finding representative of AIP. It has been reported that diffuse hypoechoic areas, diffuse enlargement, bile duct wall thickening and peripancreatic hypoechoic margins on conventional EUS are characteristic features of AIP, and the frequencies of these findings are significantly higher in AIP than in PC [107].

#### 3.4.1. Conventional EUS

Hoki N et al. [108] reported that few conventional EUS features of chronic pancreatitis (CP) were seen in patients with AIP. What is more, the frequencies of diffuse hypoechoic areas, diffuse enlargement, bile duct wall thickening and peripancreatic hypoechoic margins were significantly higher in AIP than in PC.

#### 3.4.2. CH-EUS

Hocke M et al. [109] reported that contrast-enhanced endosonography showed a unique vascularization pattern for AIP, which makes it easy to discriminate from lesions caused by PC. According to the mentioned research, lesions caused by AIP and the surrounding pancreas typically showed hypervascularization, whereas lesions caused by PC were hypovascularized [109]. Moreover, the study conducted by Ishikawa T et al. [107] CH-EUS revealed focal or diffuse iso-enhancement in most AIP cases and hypo-enhancement in most PC cases. Features of CH-EUS have been also proved to be useful in distinguishing AIP from PC in the Korean study conducted by Cho MK et al. [110]. Accordingly, it was demonstrated that, in differentiating AIP from PC, in the arterial phase of contrast agent distribution, the sensitivity and specificity of hyper- to iso-enhancement were 89% and 87%, respectively [110].

#### 3.4.3. Elastography

Dietrich CF et al. [111] in their study found that elastography of the pancreas shows a typical and unique finding with homogenous stiffness of the whole organ, and this distinguishes AIP from the circumscribed mass lesion in PDAC.

#### 3.4.4. EUS-FNA

Despite excellent results in terms of sensitivity for PC, the data are disappointing regarding the diagnosis of AIP. Previous EUS-FNA studies have reported poor to modest diagnostic performance. A prospective, a multi-centre study evaluating 50 patients with suspected AIP using a 22-gauge FNA needle reported a sensitivity of 7.9% [112]. Therefore, considering that the histological diagnosis is difficult, there is a conclusion that FNA may be used to rule out malignancy in patients with AIP. According to Matsubayashi H et al. [75], the diagnosis of pancreatic mass lesions by EUS-FNA provides a sensitivity for detecting PC tissue that exceeds 90%, making EUS-FNA the most effective tool for excluding pancreatic malignancies.

#### 3.4.5. EUS-FNB

According to the study conducted by Mizuno N et al. [113], histological diagnosis of AIP was achieved only in 37% with FNA and in all (100%) with FNB. New FNB needles, such as Franseen and Fork-tip needles [113], enabled achievement of better results in a histological diagnosis of AIP than FNA [114]. What is more, in Noguchi K et al.’s [105] study EUS-FNB was associated with a higher adverse event rate than EUS-FNA, statistically by 20%.

#### 3.4.6. Duodenal Papilla Biopsy

In 2010, Kim MH et al. [115] conducted a prospective research study to confirm the clinical validity of endoscopically accessible ampullary tissue, by evaluating IgG4 immunostaining to diagnose AIP and to distinguish it from other pancreatobiliary diseases. It confirmed the 100% specificity of positive IgG4 immunostaining of the major duodenal papilla in distinguishing AIP from pancreatobiliary malignancies.

### 3.5. Chronic Pancreatitis

CP is characterized by irreversible morphological changes, fibrosis, calcification and exocrine and endocrine insufficiency [116]. There are four modalities typically used to assess CP. The first one includes MRI. Ultrasonography (USG) is a widely and most commonly used modality for the initial diagnosis of CP. USG provides a non-invasive and cost-effective approach to evaluate the pancreas and detect structural abnormalities associated with chronic inflammation. CT is also a common imaging tool used for the initial diagnosis; however, its findings mostly appear in the advanced stages of CP, making it difficult to detect early CP. Even though, MRI allows detection of the morphological presentations of pancreatic fibrotic change, it is EUS that is believed to be the most sensitive modality for diagnosing early CP [117,118].

It is well known that advanced CP is an irreversible condition, nevertheless, Ito T. et al. [119] state that early diagnosis and intervention are crucial in managing CP and preventing further damage. By following the Clinical Practice Guidelines, healthcare professionals can implement strategies to alleviate symptoms, optimize treatment approaches and minimize complications, ultimately aiming to improve patient outcomes and potentially prevent disease progression [120]. Therefore, it is clinically crucial to diagnose CP in its early stages in order to prevent pancreatic fibrosis, progression and other complications [118]. EUS has emerged as an important imaging modality for the detection of early morphologic changes in CP/ [118]. The group of EUS experts introduced the Rosemont classification (RC) [121]. Major criteria for CP are hyperechoic foci with shadowing and MPD calculi and lobularity with honeycombing, whereas minor criteria are cysts, dilated ducts ≥3.5 mm, irregular PD contour, dilated side branches ≥1 mm, hyperechoic duct wall, strands, non-shadowing hyperechoic foci and lobularity with non-contiguous lobules [121].

EUS elastography enhances the diagnosis of CP due to its ability of measuring tissue hardness, therefore evaluation of tissue stiffness can be used to assess fibrosis of the pancreas in CP [118]. The EUS strain elastography was reported as another diagnostic method for CP, and it was shown to be correlated with the CP stages of RC [122]. However, due to its several limitations (unable to measure absolute value of hardness or is affected by the size and/or position of ROI), EUS shear-wave measurement (EUS-SWM) is a more precise tool for diagnosing CP, since it provides absolute value of pancreatic hardness [122]. A recent study conducted by Domínguez-Muñoz JE et al. [123] demonstrated that EUS-SWM was significantly positively correlated with RC stages and the number of EUS features in the RC, therefore, values obtained with EUS-SWM may reflect pancreatic fibrosis without performing histologic examinations. They came up also with data that showed the diagnostic ability of EUS-SWM for CP, the sensitivity and specificity were 100% and 94%, respectively. In 2013 Iglesias-Garcia J et al. [124] reported the sensitivity and specificity of conventional strain EUS elastography to be 91.2%, 91%, respectively. Comparing these two, EUS-SWM seems superior to conventional strain EUS elastography.

Since CP is one of the major risks for pancreatic PDAC development, it is enormously crucial to differentiate these two. Unfortunately, the diagnosis is a real challenge due to the low specificity of symptoms, imaging signs and biological markers [125]. According to Le Cosquer G et al. [125], EUS-FNAB is believed to be the technique that should provide the best information, however, its accuracy may be limited by the presence of calcifications and fibrosis of pancreatic parenchyma. To clear this limitation, artificial intelligence systems were suggested to enhance the detection of PDAC [126]. In a retrospective multi-centre study, the ability of EUS-FNAB to distinguish CP from cancer was evaluated with the sensitivity ranging from 75 to 85% and the good negative predictive value ranging from 85 to 95% [44,126,127,128]. Nonetheless, imaging of CP and PC is difficult, EUS-FNAB tends to be a helpful modality to assess the suspicious areas.

A subset of patients presents a unique diagnostic challenge as they exhibit symptoms suggestive of CP but do not show definitive abnormalities in the structure of the pancreas. These patients are commonly referred to as having early or minimal-change chronic pancreatitis (MCCP). Symptoms almost always include pain, and later exocrine pancreatic insufficiency [129]. Identifying this condition provides a distinct opportunity for early diagnosis and intervention prior to the extensive destruction of acinar cells becoming apparent on cross-sectional imaging [130]. A growing body of literature has examined alternative test to diagnose CP and MCCP—the secretin endoscopic pancreatic function test (ePFT). It detects mild exocrine dysfunction which has been considered a surrogate marker of early fibrosis [131]. In the study conducted by Albashir S et al. [132], where the patients were undergoing surgery for CP, a combined EUS with ePFT offered 100% sensitivity for detecting CP. In the recent retrospective cohort study, the ability of EUS and ePFT to predict disease progression in patients with suspected MCCP was determined [131]. The baseline ePFT result was recorded as the peak bicarbonate concentration (peak bicarbonate < 80 mmol is abnormal). Prior to collection, an intravenous dose of synthetic secretin (0.2 mcg/kg) was administered. Duodenal samples were collected at 15, 30 and 45 min after secretin stimulation and analysed for bicarbonate concentration on a hospital auto-analyser. The study found that a hazard ratio for peak bicarbonate was 4.7 for predicting future radiographic changes of CP, indicating its helpful predictive ability. To summarize EUS combined with ePFT may be helpful tests to diagnose suspected MCCP, given that they are predictive of eventual “obvious” structural changes of CP.

The sum up of the main features of different pancreatic pathologies coud be found in the Table 7.

### 3.6. Artificial Intelligence (AI)

AI is a growing field with a wide range of applications to augment the currently available modalities. AI refers to computer systems designed to imitate the human brain. Machine learning (ML) is a subset of AI that leverages vast amounts of data to identify patterns. In medical diagnostics, supervised learning methods, such as artificial neural networks (ANNs) or neural networks (NN), and support vector machines (SVM), have been investigated. Deep learning (DL), an advanced concept derived from ANN, utilizes complex layers inspired by human neurons, with convolutional neural networks (CNNs) being an example. SVM, a type of supervised ML, categorizes data based on predefined boundaries. While SVM is simpler and more generalizable than ANN, it requires significant data and development time [133].

The use of AI in EUS has the potential to enhance its diagnostic capabilities and improve the recognition of pancreatic malignancies, even in the presence of CP. In a systematic review, SVM demonstrated high sensitivity, specificity and diagnostic accuracy in distinguishing PC from CP and normal pancreas, while CNN showed slightly lower specificity. In differentiating benign and malignant IPMNs, CNN performed better than conventional EUS alone. However, the performance of AI-assisted EUS in real time and its generalizability across endoscopists of varying experience levels require further investigation. The limitations of the review include small sample sizes, retrospective designs and heterogeneity in AI methodologies.

Despite these limitations, AI outperformed conventional EUS in differentiating PC from CP and non-cancerous conditions. SVM, with its simplicity and high performance, shows promise in recognizing cancer in the presence of chronic pancreatic inflammation and in screening high-risk individuals. Prospective and real-time studies are needed to establish the role of AI in routine EUS procedures for endoscopists at all levels of training. If AI development continues to progress, it may eventually enable accurate differentiation of PC from CP and other non-cancerous conditions using EUS imaging alone, potentially revolutionizing PC screening in high-risk patients without a consensus on effective screening methods [133].

What is more AI-assisted EUS models can serve as a valuable tool for endosonographers, improving diagnostic accuracy and aiding in composite imaging for vascular staging of PCs. Additionally, AI can facilitate EUS-guided fine-needle injection for the treatment of deep pancreatic lesions. While most studies on AI are retrospective in nature, large-scale prospective clinical trials are necessary to evaluate the diagnostic accuracy of AI algorithms in real-world clinical settings. If successful, AI-assisted EUS models have the potential to become an indispensable tool in the management of patients with PC [134].

## 4. Conclusions

In summary, EUS is an indispensable tool in the diagnostic approach to gastrointestinal diseases, particularly for pancreatic conditions. Its ability to detect small lesions, differentiate various pancreatic diseases and facilitate guided interventions has revolutionized the field of gastroenterology. As technology and research progress, the future of EUS looks promising, with the potential for even greater precision and efficacy in diagnosing and managing pancreatic pathologies.

## Figures and Tables

**Figure 1 jcm-12-04630-f001:**
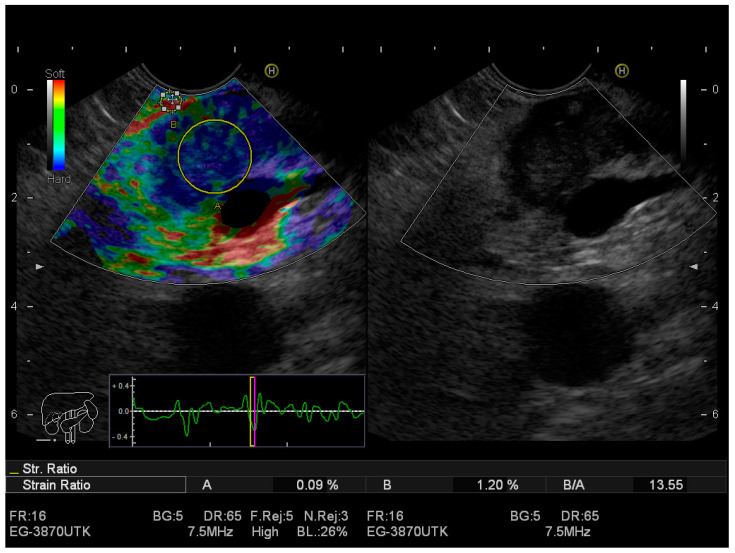
EUS–elastography; A. pancreatic adenocarcinoma adjacent to the splenic vein (yellow circle) with strain ratio on; B. without strain ratio.

**Figure 2 jcm-12-04630-f002:**
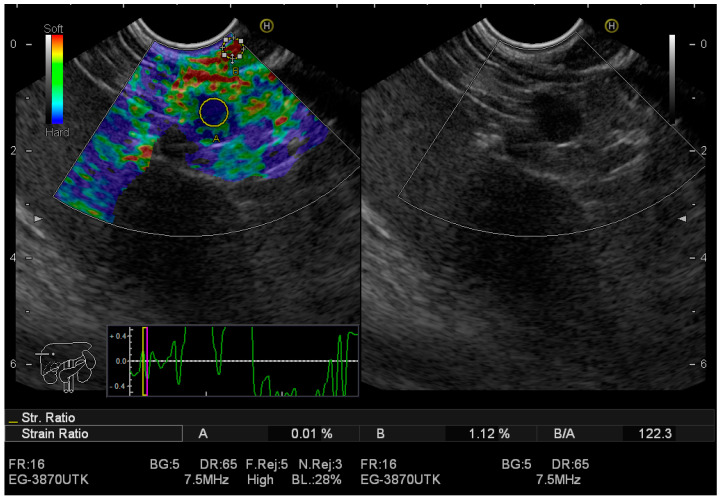
EUS elastography A. neuroendocrine tumour with high hardness strain ratio–SR = 122 is in the yellow circle); B. without strain ratio.

**Figure 3 jcm-12-04630-f003:**
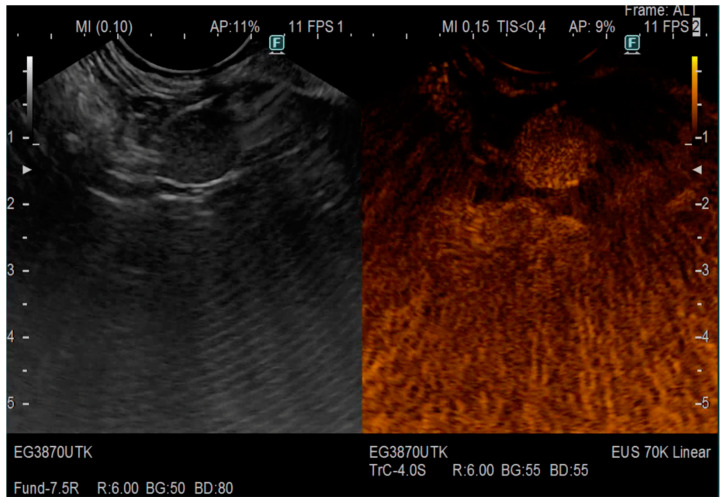
CE–EUS, NET of the pancreas, enhancing after contrast administration.

**Table 3 jcm-12-04630-t003:** Comparison of EUS-FNA and EUS-FNB of diagnostic accuracy in pancreatic cancer and SEL.

Study	Cases	Ethology	Sensitivity	Specificity	Accuracy	PPV	NPV	Comments
Kuroka N et al. [34]	94	Pancreatic cancer	78.1%	100%	81.6%	-	-	EUS-FNA
Kuroka N et al. [34]	36	Pancreatic cancer	85%	100%	85.7%	-	-	EUS-FNB
Kuroka N et al. [34]	94	SEL	100%	N/A	100%	-	-	EUS-FNA
Kuroka N et al. [34]	36	SEL	100%	N/A	100%	-	-	EUS-FNB
Oppong KW et al. [42]	108	SEL	71%	-	64%	-	-	EUS-FNA
Oppong KW et al. [42]	108	SEL	82%	-	79%	-	-	EUS-FNB
De Moura DTH et al. [43]	229	SEL	51.92%	98.39%	77.19%	96.43%	70.93%	EUS-FNA
De Moura DTH et al. [43]	229	SEL	79.41%	100%	88.03%	100%	77.78%	EUS-FNB

SEL—subepithelial lesions, N/A—not assessed, PPV—positive predictive value, NPV—negative predictive value, EUS-FNA—endoscopic ultrasound guided fine-needle aspiration, EUS-FNB—endoscopic ultrasound guided fine-needle biopsy.

**Table 4 jcm-12-04630-t004:** Meta-analysis in EUS-elastography to distinguish benign from malignant solid pancreatic masses.

Study	Cases	Sensitivity	Specificity	Diagnostic Odds Ratio	Comments
Zhang B et al. [62]	1044	95%	67%	42.28%	EUS elastography
Lu Y et al. [63]	1544 lesions	97%	67%	-	Qualitative methods
Lu Y et al. [63]	1544 lesions	97%	67%	-	Strain histograms
Lu Y et al. [63]	1544 lesions	98%	62%	-	Strain ratio

EUS—endoscopic ultrasound.

**Table 5 jcm-12-04630-t005:** Studies in CE-EUS and real-time elastography for pancreatic cancer.

Study	Cases	Ethology	Sensitivity	Specificity	Accuracy	PPV	NPV	Comments
Costache MI et al. [18]	97	Pancreatic cancer	100%	29.63%	80.41%	78.65%	100%	Real-time EUS elastography
Costache MI et al. [18]	97	Pancreatic cancer	98.57%	77.78%	92.78%	92%	95.45%	CE-EUS
Costache MI et al. [18]	97	Pancreatic cancer	98.57%	98.57%	93.81%	-	-	Combining CE-EUS and EUS elastography

EUS-endoscopic ultrasound, N/A-not assessed, PPV-positive predictive value, NPV-negative predictive value, CE-EUS-contrast enhanced endoscopic ultrasound.

**Table 6 jcm-12-04630-t006:** Studies using EUS-FNA in diagnosis of mucinous and non-mucinous pancreatic cysts.

Study	Cases	Sensitivity	Specificity	Year of the Study
Park et al. [84]	124	60%	93%	2011
Nagashio et al. [85]	68	89.2%	77.8%	2014
Okasha et al. [86]	77	73%	60%	2015

**Table 7 jcm-12-04630-t007:** Presents the main features of different pancreatic pathologies mentioned above.

Pathology	Examination	Features
Pancreatic cancer	EUS	Hypoechoic mass with irregular borders, dilatation of the proximal PD
	EUS elastography	The mean SH value (the overall hardness of a lesion) is lower than 80
	CE-EUS	Iso-enhancement or hypo-enhancement, arterial irregularity and absent venous vasculature within a mass
Chronic pancreatitis	EUS-elastography	Hyperechoic foci with shadowing and MPD calculi and lobularity with honeycombing
	CE-EUS	Hyper-enhanced lesions with preserved architecture
PNETs	CE-EUS	Hypervascularization, a low microvessel architecture
Autoimmune pancreatitis	EUS	Diffuse hypoechoic areas, diffuse enlargement, bile duct wall thickening and peripancreatic hypoechoic margins
	CE-EUS	Hypervascularization, focal or diffuse iso-enhancement
	EUS-elastography	Homogeneous stiffness of the whole organ

EUS—endoscopic ultrasound, CE-EUS—contrast-enhanced endoscopic ultrasound, PNETs—pancreatic neuroendocrine tumours.

## Data Availability

Not applicable.

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
