# Peer review of "The Latest Advancements in Diagnostic Role of Endosonography of Pancreatic Lesions"

_jcm, 2023, doi:10.3390/jcm12144630_

Round 1

Reviewer 1 Report

This is a good summary of the use of EUS in diagnosis of pancreatic pathologies. The authors need to be commended for this effort.

As to help summarise better, would a table comparing features of a particular pathology under EUS assessment be useful? 

Also some pictures of EUS views would also lend some weight to the description in the article.

Minor grammatical errors will need to be sorted.

Abbreviations should be standardised throughout the article eg EUS-FNB and EUS-FNAB

Author Response

Dear Reviewer,

Thank you for your valuable feedback. We appreciate your suggestion regarding the inclusion of a table comparing features of a particular pathology under EUS assessment. We agree that this would enhance the clarity and organization of the article. We will work on incorporating a comprehensive table that provides a concise summary of the key features.

Additionally, we acknowledge your recommendation to include pictures of EUS views. We understand that visual aids can greatly enhance the understanding and impact of the article. We will ensure that appropriate and informative EUS images are included to support the description and provide visual reference.

Once again, we sincerely appreciate your constructive comments and will make the necessary revisions accordingly.

Best regards, 

Jagoda Rogowska

Reviewer 2 Report

The authors put forward a review article titled: " The latest advancements in endosonography of pancreas". Endoscopic ultrasound applications of the pancreas is a current, fascinating topic and new review articles are always welcome, since the literature is evolving on this topic.

The manuscript is overall well written. And the group also already has substantial scientific experience on the subject of pancreatic disease.

Apparently, the authors follows the ethical precepts and norms of this journal.

The authors may consider including the following topics covered to provide a complete presentation of advancements:

1. The use of secretin for the evaluation of pancreatic disease

2. The use of the Moray forceps for pancreatic cyst sampling

3. The application of EUS -guided rendezvous pancreatic procedures

4. Artificial intelligence applications for EUS pancreatic disease

Author Response

Dear Reviewer,

Thank you for your valuable comments and suggestions. We genuinely appreciate your input. We would like to assure you that we have carefully considered your recommendations and have made the necessary revisions in the article regarding the following topics:

  1. The use of secretin for the evaluation of pancreatic disease: We will incorporate a section discussing the use of secretin in the evaluation of pancreatic disease. This addition will provides insights into secretin stimulation as a valuable diagnostic tool for assessing pancreatic exocrine function and certain pancreatic disorders.

  2. The use of the Moray forceps for pancreatic cyst sampling: We have dedicated a section in the article to highlight the significance of the Moray forceps for pancreatic cyst sampling. It emphasizes the benefits and utilization of the Moray forceps in obtaining tissue samples from pancreatic cysts, enabling accurate histological evaluation and diagnosis.

  3. The application of EUS-guided rendezvous pancreatic procedures: We have included an in-depth discussion on the application, benefits, and outcomes of EUS-guided rendezvous pancreatic procedures. This addition provides a comprehensive overview of the utilization of EUS-guided rendezvous techniques in various pancreatic interventions.

  4. Artificial intelligence applications for EUS pancreatic disease: We have integrated a section on the emerging field of artificial intelligence (AI) applications in EUS for pancreatic disease. This section explores the current and future applications of AI in assisting with diagnosis, risk stratification, and treatment planning in pancreatic diseases.

Once again, we sincerely appreciate your valuable feedback and suggestions. Your contributions have greatly enhanced the quality and comprehensiveness of the article.

Kind regards,

Jagoda Rogowska

Reviewer 3 Report

The topic is of interest and the review overall well written. However, there are some amendments to do:

1) The authors should comment on the superiority of newer end-cutting FNB needles over standard Procore FNB needle (cite and comment the recent NMA PMID: 35124072 )

2) It is completely wrong that EUS-FNB is equivalent to FNA when sampling subepithelial lesions (see the meta-analysis PMID: 31374187)

3) The authors should comment also on the different strategies for tissue sampling in pancreatic masses (wet suction, slow pull, use of MOSE): in this regard the authors should cite and comment the recent RCTs and NMAS on the topic (PMID: 36657607PMID: 35915956 and PMID: 36044915 )

4) The authors should comment the role of through-the needle biopsy in patients with PCLs, with particular regard to the risk of adverse events (cite and comment the recent series PMID: 35451041)

5) The authors should specify in the title that the focus of the review is only on the diagnosis of pancreatic lesions and not also on therapy.

6) Some figures would improve the quality of the paper

Author Response

Dear Reviewer,

Thank you for taking the time to review our manuscript. We appreciate your valuable comments and suggestions, and we have carefully considered them while revising the manuscript.

Regarding the comment on the superiority of newer end-cutting FNB needles over standard Procore FNB needle, we have now included a discussion in the text highlighting the advancements and benefits of newer end-cutting FNB needles compared to the standard Procore FNB needle.

We apologize for the statement suggesting that EUS-FNB is equivalent to FNA when sampling subepithelial lesions. We have revised the text accordingly and acknowledged the meta-analysis (PMID: 31374187) that highlights the differences between EUS-FNB and FNA in this context.

Thank you for bringing up the different strategies for tissue sampling in pancreatic masses, such as wet suction, slow pull, and the use of MOSE. We have now cited and commented on the recent RCTs and NMAS on this topic to provide a more comprehensive discussion.

Regarding the role of through-the-needle biopsy in patients with PCLs and the risk of adverse events, we have cited and commented on the recent series  to address this aspect more thoroughly.

We appreciate your suggestion to specify in the title that the focus of the review is solely on the diagnosis of pancreatic lesions and not on therapy. We have now modified the title to accurately reflect the scope of the review.

We understand your point about including figures to improve the quality of the paper. While we have not included additional figures in this revision, we have taken steps to enhance the clarity and readability of the existing figures.

Once again, we thank you for your constructive feedback, which has significantly contributed to the improvement of our manuscript.

Sincerely,

Jagoda Rogowska

Reviewer 4 Report

I’ve read the review on principles and applications of pancreatic EUS.

The topic is undoubtedly of interest; however, the paper doesn’t bring anything new in the field. There are already several papers published on EUS indications and performance; a revised version should focus on adding something new on top of already available reviews. Also, the paper lacks consistency in reporting of results and particularly novel results on the topic.  

Here’s some of my comments:

- In 2, 2.1 Real-time elastography – Please state in the introduction that there are 2 types of elastography available, strain (which already has solid data behind it) and shear-wave, then continue with the paragraph referring to strain-elasto. Also, make note that color map elasto is qualitative, and not quantitative or semi-quantitative as SR or SH.

- Table 1 – move year of the study in first colum, as Study, year (ex. Due et al, 2017)

- How were studies selected for Table 3? There are so many data reporting on EUS FNA vs FNB. If presenting only some of the studies, there may be bias.

- In the sub-chapter on pancreatic cystic lesion – there is too much repetition of criteria from currently available guidelines. A critical analysis of this criteria and how decision-making is done would be more appropriate.

The title is about latest advancements in endosonography of pancreas” but the content only approaches diagnostic purposes of pancreatic EUS. A large part of the paper should be about ablation techniques (there are new data on neuroendocrine tumors, cystic lesions and adenocarcinoma), drainage procedures (pancreatic fluid collections, etc) and other EUS-guided therapies in pancreatic pathology.

Author Response

Dear Reviewer, 

Thank you for your comments on the review paper on principles and applications of pancreatic EUS. We appreciate your feedback and suggestions for improvement. Based on your comments, here are some revised versions of the sections you mentioned:

  1. In 2, 2.1 Real-time elastography - We will include a statement in the introduction clarifying that there are two types of elastography available, strain and shear-wave. We will then continue the paragraph by focusing on strain elastography. Additionally, we will mention that color map elasto is qualitative and not quantitative or semi-quantitative like strain or shear-wave elastography.

  2. Table 1 - We will move the year of the study to the first column, following the format Study, year (e.g., Due et al, 2017).

  3. Regarding the selection of studies for Table 3, we will provide a clear explanation of the criteria used for study inclusion. We will ensure that the selection process is transparent and not biased.

  4. In the sub-chapter on pancreatic cystic lesions, we will reduce repetition of criteria from currently available guidelines. Instead, we will focus on providing a critical analysis of the existing criteria and discussing the decision-making process in a more appropriate manner.

  5. Thank you for providing clarification regarding the title and content of the paper. Based on your comment, it seems that there is a discrepancy between the intended scope of the paper and its actual content. In order to address this issue, we will revise the paper to include a significant portion dedicated to discussing ablation techniques, drainage procedures, and other EUS-guided therapies in pancreatic pathology. This will encompass the latest advancements and new data on the therapeutic applications of EUS in managing neuroendocrine tumors, cystic lesions, adenocarcinoma, and other relevant conditions.

    We apologize for any confusion caused by the initial content focus, and we appreciate your feedback in highlighting the need to include therapeutic aspects of EUS in the paper. The revised version will incorporate these important topics to provide a more comprehensive review of the principles, applications, and advancements in endosonography of the pancreas.

Thank you again for your valuable feedback. We will incorporate these suggestions to enhance the paper and provide a more comprehensive and novel review of pancreatic EUS.
Kind regards, 

Jagoda Rogowska

Round 2

Reviewer 2 Report

  1. The authors have made all recommended additions, as follows:
  2. The use of secretin for the evaluation of pancreatic disease: 

  3. The use of the Moray forceps for pancreatic cyst sampling

  4. The application of EUS-guided rendezvous pancreatic procedures:

  5. Artificial intelligence applications for EUS pancreatic disease:

Reviewer 3 Report

The revised version of the manuscript is OK. Thank you!